# Feed Supplementation Detection during the Last Productive Stage of the Acorn-Fed Iberian Pig through a Faecal Volatilome Analysis

**DOI:** 10.3390/ani13020226

**Published:** 2023-01-07

**Authors:** Pablo Rodríguez-Hernández, María José Cardador, Rocío Ríos-Reina, João Simões, Lourdes Arce, Vicente Rodríguez-Estévez

**Affiliations:** 1Department of Animal Production, UIC ENZOEM, International Agrifood Campus of Excellence (ceiA3), University of Córdoba, Campus de Rabanales, 14071 Córdoba, Spain; 2Department of Analytical Chemistry, Institute of Fine Chemistry and Nanochemistry, Marie Curie Annex Building, International Agrifood Campus of Excellence (ceiA3), University of Córdoba, Campus de Rabanales, 14071 Córdoba, Spain; 3Área de Nutrición y Bromatología, Departamento de Nutrición y Bromatología, Toxicología y Medicina Legal, Facultad de Farmacia, Universidad de Sevilla, C/P. García González n°2, 41012 Sevilla, Spain; 4Veterinary and Animal Research Centre (CECAV), Associate Laboratory for Animal and Veterinary Sciences (AL4AnimalS), Department of Veterinary Sciences, School of Agricultural and Veterinary Sciences, University of Trás-os-Montes and Alto Douro (UTAD), 5000-801 Vila Real, Portugal

**Keywords:** pasture-raised, faeces, volatilome, feed supplementation, grazing diet

## Abstract

**Simple Summary:**

The acorn-fed Iberian pig is known worldwide due to its unique and exclusive feeding regime for finishing, which consists exclusively of grazing acorns and grass in the *dehesa* agroforestry system. However, some farmers try to increase the carrying capacity of their farms using supplementary feed for grazing pig fattening. The current regulation to certificate products using this breed is based on traceability and visual observations, which can be considered subjective. Therefore, considering that background, the present study seeks to provide an analytical approach through the analysis of faecal volatilome to authenticate the acorn-fed Iberian pig diet and to discriminate the animals that receive supplementary feed during the finishing period.

**Abstract:**

The acorn-fed Iberian pig is known worldwide due to the quality of the resulting products commercialized after a natural and free grazing period of fattening in the *dehesa* agroforestry ecosystem. The quality regulation of the pig breed reserves “acorn” denomination for only those products obtained from animals exclusively fed grazing acorns and other natural resources; however, sometimes, feed supplementation of the pig’s diet is fraudulently employed to reach an earlier slaughtering weight and to increase pig stocking rate, a strategy called *postre* (meaning “feed supplement”). In this sense, although many studies focused on Iberian pig diet have been published, the field detection of feed use for acorn-fed pig during the last finishing stage foraging in the *dehesa*, a practice which clashes with the official regulation, has not been explored yet. The present study employs a volatilome analysis (gas chromatography coupled to ion mobility spectrometry) of a non-invasive biological sample (faeces) to discriminate the grazing diet of only natural resources, that acorn-fed Iberian pigs are supposed to have, from those pigs that are also supplemented with feed. The results obtained show the suitability of the methodology used and the usefulness of the information obtained from faeces samples to discriminate and detect the fraudulent use of feed for acorn-fed Iberian pig fattening: a classification success ranging between 86.4% and 100% was obtained for the two chemometric approaches evaluated. These, together with the results of discriminant models, are discussed, in addition to the importance that the methodology optimized implies for the Iberian pig sector and market, which is also introduced. This methodology could be adapted to control organic farming animals or other upstanding livestock production systems which are supposed to be fully dependent on a natural grazing diet.

## 1. Introduction

Nowadays, there is an increasing number of consumers willing to pay a premium for a pasture-raised attribute even on top of an organic price premium [1]. Iberian pig is an autochthonous breed only reared in southwestern Iberian Peninsula (Portugal and Spain). This breed holds an important international reputation due to the quality of its products, which are consumed in many countries worldwide. Iberian pig has traditionally been linked to the ecosystem called *dehesa* in Spain and *montado* in Portugal, characterized by a high environmental value [2,3]. The traditional finishing of this breed based on free grazing of acorns is considered an upstanding system for animal welfare, natural diet and environmental conservation [4]. Nevertheless, nowadays, there are different productive systems in terms of feeding regime and animal freedom defined by the Spanish regulation (Real Decreto 4/2014) [5]. In this sense, it is possible to find a range from totally intensive farms where the animal feeding is based on an intensive diet with concentrate feed to extensive systems in *dehesa* farms where pigs freely graze during the last stage of the fattening period (a minimum of 2 months to gain a minimum of 46 kg of body weight), depending on acorns, grass, and other natural resources [3,4]. This period is known as the *montanera* fattening period (meaning “pannage”).

The latter are known as acorn-fed Iberian pigs, whose products are the most popular of the whole Iberian sector. This is due not only to its excellent organoleptic and nutritional profile, which is considered to be of superior quality to those from intensive systems [6], but also to the added value of the freedom conditions and the extensive systems where the animals are reared [7,8]. These attributes are increasingly appreciated by the consumers [1]. The official regulation currently available for Iberian pig products, Real Decreto 4/2014, reserves the term “acorn” for only “products from animals slaughtered immediately after the exclusive feeding of acorns, grass and other natural resources of the *dehesa*, without the contribution of supplementary feed” [5]. Therefore, the animal feeding must be based only on a natural and grazing diet in the case of “acorn” denomination. The regulation also considers another two commercial categories depending on the type of feeding regime during finishing: the “Cebo” (meaning feed-fed) and “Cebo de Campo” (meaning outdoor feed-fed) denominations. However, the final price on the market as well as the added value of both categories are substantially lower than the “acorn” Iberian pig products.

There is an irregular feeding strategy called *postre* (meaning “feed supplement”), used during the last finishing stage, while foraging acorns, in *dehesa* farms (where acorn-fed pigs are reared) which clashes with the basis of the above-mentioned “acorn” denomination established by the Spanish regulation. That feed supplement consists of complementing animals’ diet, which should be based only on foraging natural resources (basically acorns and grass) [3]. The aim of this supplementation is to reach the slaughtering weight earlier, implying a lower dependence on natural resources and foraging, and allowing a higher stocking rate of finishing pigs, which should be ≤1 pig ha^−1^ [9]. There is some controversy about this practice because, although defended by some farmers as a handling measurement which allows animal monitoring and control in free-range *dehesa* farms (extensive systems), it is not covered by the “acorn” category in Real Decreto 4/2014 [5]. Furthermore, the final products are commercialized worldwide as “acorn” products without any mention of this practice but obtaining the same economic value; the consequence of using *postre*, if discovered during field inspections, would be a change in commercial category and a lower price.

Although the regulation (Real Decreto 4/2014) establishes mandatory inspection and certification visits to ensure that feeding and rearing conditions conform to the regulation [5], these are only based on visual observations made by technicians, without using any objective analytical method or procedure. Therefore, the detection of inappropriate practices such as *postre* supplement may be difficult, especially in large *dehesa* farms.

Against that background, the aim of this work was to evaluate an objective method to detect the use of supplementary feed or *postre* in the final productive stage of acorn-fed Iberian pigs. Considering the great deal of interesting information that faecal volatilome analysis has shown to provide in different fields of animal science, such as animal reproduction, feeding or health [10,11,12] as well as the sampling facility of this type of sample [13], it is proposed to evaluate volatile compounds (volatilome) from faeces through gas chromatography coupled to ion mobility spectrometry (GC–IMS). This technique has been successfully employed to authenticate Iberian pig production analyzing different samples: raw subcutaneous fat [14], dry cured Iberian ham slices [15] and fat from final products [16]. Recently, faeces have been also used in order to discriminate the three current categories defined by the Spanish regulation [13], but not focusing on acorn denomination and *postre* practice as the present study does.

The proposed analysis of faeces with GC–IMS may help technicians during inspection visits and contribute to the authentication of the feeding regime given to the acorn-fed Iberian pig in *montanera*, using a non-invasive and animal welfare-friendly sampling method. What is more, this methodology to authenticate pasture-raised animals could be useful for other upstanding livestock production systems.

## 2. Materials

### 2.1. Samples

A total of 75 individual fresh faeces samples (from 75 different pigs) were collected in six Iberian pig extensive *dehesa* farms. All farms were certified by the Spanish administration and therefore were producers of animals labelled as acorn fed according to the regulation [5]. However, two groups were differentiated among the farms included in this study in terms of the truthful animal feeding regime: four farms with a completely extensive diet based on grazed acorns, grass and natural resources (acorn-fed group), and two farms with the above-mentioned extensive diet and feed supplementation (supplementation group). The adaptation time of animals to the feeding regimes evaluated was at least 1.5 months before faeces sampling. All the animals included in this study were similar between the two groups: these were male and female castrated Iberian pigs in the last fattening stage, approximately from 100 to 160 kg (kg) live weight and with an average age of approximately 18 months at sampling. The initial animal live weight ranged between 92 and 115 kg, obtaining a minimum weight gain of 46 kg during at least 60 *montanera* days. The two groups compared were mixed in terms of animal sex to avoid a possible influence of this variable on discrimination results. Considering that the use of *postre* is a controversial practice, the search for this type of farm was difficult and thus fewer animals were sampled in this group; hence, 47 samples were obtained in the first group of farms, while the remaining 28 were collected in farms where pigs had feed supplementation. According to farmers’ information, supplemented pigs received approximately 1 kg of feed per day. The main components of the feed employed (*postre*) in farms were cereals, oil seeds, minerals, animal fat and vegetal oil, and the analytical composition was: 10.9–14.02% crude protein, 3.12–7.64% crude fibre, 3.93–7.37% crude fat, 4.10–7.99% crude ash, 0.54–0.70% lysine, 0.16–0.24% methionine, 0.40–1.52% calcium, 0.35–0.66% phosphorus, and 0.13–0.26% sodium.

Sample collection was carried out immediately after spontaneous defecation while pigs foraged. This process ensured that each animal was sampled only once. As animals were freely grazing and their immobilization was not feasible, faeces samples were collected directly from the ground. The parts of samples in contact with soil were discarded to avoid contamination. Afterwards, faeces samples were refrigerated, transported to the laboratory and stored at −18 °C.

Faeces samples were thawed overnight and processed the same day of analysis to prevent degradations: 1.5 g of fresh faeces was weighted with a precision balance in a 20 mL glass vial, which was hermetically closed using a metallic cap and a silicone septum.

### 2.2. Instrumentation and Software

Volatile organic compounds from faeces were determined using a GC–IMS instrument (FlavourSpec^®^, G.A.S., Dortmund, Germany) equipped with an autosampler (Combi Pal, CTC Analytics AG, Zwingen, Switzerland). The device was equipped with a fused silica 30 m × 0.32 mm ID capillary column with a non-polar 0.5 μm HP-5 stationary phase (5%-Phenyl)-methylpolysiloxane (Agilent, Santa Clara, CA, USA). IMS instrument data were acquired in the positive mode and evaluated using LAV^®^ 2.1.1 software supplied by G.A.S.

The GC–IMS method employed for faeces analysis was based on a previous study published [13]. In brief, 1.5 g of each individual sample was carefully weighed into a 20 mL headspace vial and then incubated at 60 °C for 15 min with constant stirring. After incubation, 100 μL headspace was injected using an 80 °C heated syringe (2.5 mL Hamilton syringe with a 51 mm needle) into the capillary column heated at 45 °C in the splitless mode. Nitrogen (99.999% purity) was used as the carrier gas with a flow ramp of 1–20 mL min^−1^. Analytes were driven to the ionization chamber by a tritium ionization source in the positive ion mode. The 9.8 cm drift tube was operated at 75 °C and nitrogen was also used as drift gas at 150 mL min^−1^.

PLS Toolbox software v.6.2 (Eigenvector Research, Manson, WA, USA) working under MATLAB^®^ environment (The Mathworks Inc., Natick, MA, USA, 2007) was employed for data treatment and chemometric multivariate analysis. SPSS^®^ statistical software package version 22.0 (IBM Corporation, Somers, NY, USA) was used to evaluate differences and possible tendencies between the data obtained after GC–IMS analysis of faeces samples through a Student t-test. Those *p* values below 0.05 were considered significative. The experimental unit along the discriminant models was the pig (the individual faeces sample collected from each animal).

### 2.3. Data Analysis: Chemometrics

Chemometric techniques were carried out to treat and process the huge amount of data generated after GC–IMS analysis. Data processing was carried out through two approaches, evaluating the whole spectral fingerprint (non-targeted approach; Model 1), and also processing individual features (targeted approach; Model 2) to carry out a classification of faeces samples in two categories or groups (two feeding regimes): extensive diet based on acorn and natural resources (acorn-fed group) vs. supplementation in addition to the above-mentioned extensive diet (supplementation group). All spectra obtained were aligned before data treatment employing a specific feature present in every sample as reference.

For the non-targeted approach, raw IMS data were converted to .csv format with LAV^®^ software and data pretreatment consisted on the normalization of IMS data using the reactant ion peak (RIP) intensity value of each sample, followed to a smoothing procedure based on Savitzky–Golay filtering and a baseline correction and data reduction by discarding the parts where no relevant information was presented and considering only the zone which contains the signals: from 1150 to 2150 ms of drift time and from 196 to 1690 s of retention time. Then, the spectra selected for each sample were arranged consecutively in a single data row by spectra concatenation, obtaining the final matrix that was preprocessed by mean centering prior to model building by partial least squares-discriminant analysis (PLS-DA) (Model 1).

The targeted strategy consisted of the selection of 409 specific features instead of the whole fingerprint to carry out discriminant analysis. LAV^®^ software was used to obtain feature intensity of each signal, and afterwards this matrix was employed for developing the classification model (Model 2) based on a PLS-DA. The pretreatment used in this approach after modelling consisted of autoscaling (mean-centering and scaling to unit variance). The variable importance for the projection (VIP) of each feature was also obtained in order to evaluate its influence on sample classification as well as to reduce data complexity and variables with a limited significance for model classification.

The chemometric models developed for the non-targeted and targeted approaches employed 70% of samples as the training set (calibration set) to calibrate models and the remaining 30% was used as the validation set or blind samples (prediction set). The number of latent variables (LVs) for PLS-DA models in each approach was selected considering the root mean square error of the cross-validation and the prediction. A principal component analysis (PCA) was also performed in both strategies before PLS-DA models to explore data and detect possible outliers.

The classification ability of the two discriminant models designed in both approaches (targeted and non-targeted) was assessed through the evaluation of the figures of merit for calibration, cross-validation and prediction: classification error (samples correctly classified vs. total samples), and sensitivity (SE) and specificity (SP) according to the following equations:$$\mathrm{Sensitivity}\;\left(\%\right)=\left[\frac{\mathrm{TP}}{\mathrm{TP}+\mathrm{FN}}\right]\times 100\%$$
$$\mathrm{Specificity}\;\left(\%\right)=\left[\frac{\mathrm{TN}}{\mathrm{TN}+\mathrm{FP}}\right]\times 100\%$$
where TP and TN represent the number of samples correctly classified (TP: number of acorn-fed group samples classified inside this class; and TN: number of supplementation group samples predicted as supplementation). On the other hand, FP and FN represent the number of samples misclassified (FP: acorn-fed group samples predicted to this class; FN: supplementation group samples assigned to supplementation class).

## 3. Results

A comparison between GC–IMS plots of the two feeding regimes evaluated in the present study is available in Figure 1. In this figure, individual features employed in Model 2 can be noticed (as red signals), which are all characterized by specific drift and retention times (X and Y axes, respectively). Some differences can be visually addressed in the two GC–IMS plots of Figure 1, so chemometrics was employed to evaluate this preliminary visual discrimination.

### 3.1. Chemometric Models

PCA revealed no outliers for spectral fingerprinting (Model 1) and individual feature (Model 2) approaches; therefore, both models were carried out using all the samples analyzed with GC–IMS. Although classification results were satisfactory in both models, these were different depending on the approach employed for data treatment: a higher classification success was obtained in Model 2, which achieved 100% of correct prediction, in comparison with Model 1, with an 86.4%. Table 1 and Table 2 show the confusion matrices obtained after external validation: while all the samples were successfully classified in Model 2, three sample of the validation set of Model 1 were misclassified. Those samples of Model 1 corresponded to the acorn-fed group, which were incorrectly classified in the supplementation group (Table 1).

The statistical assessment of the performance achieved by discriminant models was carried out through the comparison of classification error, SE and SP parameters, for calibration, cross-validation and prediction. These results are available in Table 3. In general, a greater performance was obtained by Model 2 for calibration, cross-validation, and prediction in comparison with Model 1. The maximum error classification obtained for the individual features approach (Model 2) was only 5%; and SE and SP values were always above 95%. On the other hand, fingerprint approach (Model 1) achieved a lower statistical performance, with a classification error above 15% in both calibration as well as in cross-validation and prediction; and its SE and SP percentages ranged between 75 and 100%.

The differences found between both models in terms of statistical performance can be also noticed on the score plots of the models, which are shown in Figure 2. Some overlapping of the two groups evaluated is clearly visible and higher in Figure 2A (Model 1) in comparison with Figure 2B (Model 2), where the two categories are more separated and, hence, obtaining greater percentages of classification error, SE and SP.

In addition, differences could be also observed in Figure 2 between the two group of samples evaluated in both models (Figure 2A,B). The distribution area of points belonging to acorn-fed group was higher than in supplementation group, where points tended to be more concentrated. Although this difference can be noticed in the score plots of both models, it is more marked in Figure 2A (Model 1).

### 3.2. Evaluation of Variables with Importance in Prediction and Feature Differences for the Targeted Approach (Individual Features)

Considering the large number of initial features or variables available after faeces analysis with GC–IMS (409 features for the targeted approach), VIP scores were studied in order to reduce data complexity and simplify discriminant models. Moreover, the objective of this strategy was to evaluate how many variables or features could be removed from the model without reducing the discriminant and statistical performance achieved. Thus, those features with a VIP value lower than 1 in the first model (Model 2) were eliminated and the model was performed again, but only with those features with VIP scores higher than 1. This process of feature or variable reduction using VIP scores was repeated three times, hence obtaining four different PLS-DA models. The three different evaluations of VIP scores carried out are available in Figure 3, where the dashed red line shows the threshold beyond which features have a VIP score higher than 1. As can be seen in the previous figure, there was a substantial number of variables with VIP values lower than 1 when VIP scores were studied in the three cases.

The resulting loading plots of the four models performed for the targeted approach after VIP evaluation are available in Figure 4: Figure 4A is the first PLS-DA model performed with all the 409 initial features; Figure 4B,C constitute the second and third discriminant models obtained after feature (variable) removal through VIP evaluation of Figure 3A,B, respectively; and, lastly, Figure 4D corresponds to the final PLS-DA model performed once the features of Figure 3C with VIPs lower than 1 were removed. Thus, the initial number of features of the PLS-DA model performed with a targeted approach was greatly reduced; while the first model (Figure 4A) was obtained using 409 features or variables, the fourth one (Figure 4D) was performed only with 21 variables, which constitute 5.1% of the initial number. These latter variables are the same 21 which show a VIP value higher than 1 in Figure 3C.

Despite the important reduction in features, the statistical performance of the resulting model was not affected: the fourth PLS-DA model performed with only 21 variables achieved the same validation success (100%) as the first model but using 388 fewer features.

SPSS^®^ statistical software was employed to perform a Student *t*-test to evaluate the significance and tendencies between the 21 feature intensities used for the final PLS-DA model obtained. These results are available in Table 4. Significative differences between acorn-fed and supplementation groups were detected in 18 out of the 21 total features (*p* < 0.05), obtaining *p* values below 0.001 in 11 of them. The intensity obtained for the final selection of 21 features was generally higher in supplementation than in the acorn-fed group. Only in five of them a different tendency was observed, whose intensity was higher for the acorn-fed group; however, significative differences were only detected in two of these features (Table 4).

## 4. Discussion

To date, there is no field study available focused on the use of *postre* for Iberian pig feeding and its differentiation from the exclusive and extensive diet that the acorn-fed Iberian pig is supposed to have. However, an approximation of faecal volatilome usefulness to discriminate through GC–IMS analysis the Iberian pig diet according to the current commercial categories defined by the regulation for Iberian pig production [5] was recently published [13]. The above study was exclusively focused on the current commercial categories available at the market, though some information about *postre* practice was also shared; after in-depth investigation of the origin of the samples included in that study, some of those belonging to acorn-fed pigs which were misclassified by chemometric models were demonstrated to proceed from farms where *postre* practice was employed for animal’s feeding, albeit an specific evaluation of GC–IMS potential to detect *postre* use was not performed. Therefore, considering the above-mentioned possible influence of *postre* on chemometric model’s discrimination, volatilome analysis through GC–IMS could also constitute a possible methodology capable of distinguishing the use of that strategy in acorn-fed Iberian pig feeding, apart from the differentiation between the official commercial categories currently defined by the regulation, as Rodríguez-Hernández et al. (2022), previously demonstrated.

### 4.1. Chemometric Models

Although the misclassified samples of Model 2 belonged to the acorn-fed group, this finding is considered quite interesting, as every supplementation sample was correctly predicted by the discriminant model; thus, although the model failed in acorn-fed group prediction and some false negatives could be obtained for this category, considering the accuracy and success detected for the supplementation group (100%), chemometric tools may constitute a feasible option to clearly detect the use of feed supplementation in this type of system called *montanera*, because every supplementation sample was correctly classified as such. On the other hand, the classification error of Model 1 could be related to the heterogeneity of acorn-fed pig diet, because these animals freely graze in the *dehesa*, existing different dietary preferences depending on the individual, mainly showed in the acorn and grass proportions grazed [9] and, even, the different composition of acorns grazed along the montanera or mast season [3]. In this sense, it has been demonstrated that the acorn-fed Iberian pig can eat up to 16 different minor resources during *montanera* fattening period, thus giving rise to important differences between animal feeding even in the same *dehesa* farm [3]. Against that background, in order to improve the classification results of the non-targeted approach (fingerprint), as shown in Model 1, a higher number of samples and farms would be necessary, trying to cover as much variability as possible.

The higher distribution area of points noticed in both models for the acorn-fed group in comparison with the supplementation group (Figure 2) could be explained considering the above-mentioned diet heterogeneity of acorn-fed pigs, which are fed with several natural resources in the *dehesa* [3]. In contrast, the use of a considerable amount of supplementary feed or *postre* for acorn-fed pigs’ feeding seems to increase the homogeneity of samples, which can be noticed on the more concentrated score plots belonging to the supplementation group in Figure 2. According to Rodríguez-Estévez et al. (2010), Iberian pigs have a mean daily ingestion of acorns and grass dry matter of 3.4 ± 0.14 kg over *montanera* season; therefore, 1 kg of feed per day would be approximately 30% of the daily intake, using a formulated feed specially prepared to imitate the acorn and grass diet, and the corresponding fatty acid profile from free grazing animals, which is required and highly appreciated by dry cured ham industries [17]. Hence, the use of the feeding strategy known as *postre* could reduce the animal diet heterogeneity and, consequently, would homogenize the volatilome profile of faeces samples enough to discriminate acorn-fed Iberian pigs from those which are supplemented with feed (supplementation group). These results are consistent with the influence observed of Iberian pig feeding regime on the volatilome composition of different matrices such as cured ham slices [15], raw fat [14,17], cured fat [16] or, recently, faeces [13].

The greater classification results obtained with a targeted approach in comparison with a non-targeted one are also consistent with previous studies focused on Iberian pig diet and GC–IMS. In this sense, Arroyo-Manzanares et al. (2018) employed this methodology to authenticate the feeding regime of the Iberian pig using cured ham slices, obtaining very similar results than the ones presented in this study: while a classification rate of 90% was obtained for spectral fingerprint approach (non-targeted), the use of individual features (targeted) allowed a classification rate of 100%. Nowadays, there is a common discussion about the use of these two approaches, from which different advantages and disadvantages can be addressed: a non-targeted option requires powerful computation and chemometric knowledge, obtaining a more complex validation and implementation [18]. However, as it is rapid, reliable and less problematic than a targeted one, any early implementation, validation difficulties and costs could be justified [19]. On the other hand, the selection and use of specific features are considered a hard task, as there may also be some interpersonal variability; this is further accentuated when a complex matrix is studied. Although that selection removes unnecessary information, compound identification and its implementation in official methods of analysis in the near future are considered by some authors much easier in comparison with a non-targeted option [19].

### 4.2. Evaluation of Variables with Importance in Prediction and Feature Differences for the Targeted Approach (Individual Features)

The study of VIPs allows the evaluation of the influence of individual variables in a PLS/PLS-DA model; in chemometrics, VIP scores could be used for variable selection, as those variables with values higher than 1 are considered to have a significative influence, while those ones with values lower than 1 could be excluded from the models due to its scarce relevance [20,21]. This parameter is often used for the analysis of data such as metabolome or volatilome, in order to select features from multiple potential compounds which could be considered as important biomarkers [21].

Although 388 out of 409 variables were removed for Model 2, the classification success remained the same (100%). This finding highlights the importance that a correct selection of features has for the statistical or discriminant performance of a PLS-DA model using individual compounds (targeted approach). The inclusion of a large number of features in discriminant models involves a tedious work based on visual inspections of GC–IMS topographic plots. However, good rates of classification can be also reached with a lesser number of features if they are properly chosen. This is consistent with the reduction in over 380 features in the present study without affecting the validation success of the PLS-DA model. The score plot available in Figure 2B corresponds to the fourth PLS-DA model performed only using the information of 21 features, after evaluating VIP values and reducing features three times. The number of final features (21) included in the model could not be more reduced because, thereafter, its statistical performance was greatly affected.

VIP evaluation has been employed in studies focused on the discrimination or classification of livestock feeding regimes. Recently, Freire et al. (2021) included VIP evaluation between the different approaches used in their study to differentiate the Iberian pig diet through GC–IMS analysis of ham samples. On the other hand, [22] studied two metabolomic methodologies in order to identify the effect of different feeding regimes on lamb meat, selecting those metabolites with a VIP value exceeding 1 as differential metabolites or potential biomarkers.

Considering the results obtained using the Student t-test, a possible biomarker or indicator for feed supplementation of Iberian pig diet would be more feasible to obtain instead of a marker of acorn-fed pigs, since in 16 out of the final 21 features used in Model 2, a higher intensity was detected in this group than in acorn-fed one (Table 4). However, the identification of the above-mentioned features was not carried out because it is considered risky by the authors; considering that this study constitutes the first evaluation of the use of an analytical methodology to detect the use of feed supplementation during the last stage of the acorn-fed Iberian pig fattening, the publication of potential biomarkers associated to the use of *postre* or to the extensive diet in *montanera* could be used to design specific feeds that conceal the fraud. In fact, a similar fact was detected a few years ago when the classification of Iberian products was based on the determination of four fatty acids: although a chromatograph was employed for this determination, the publication of these specific fatty acids allowed the feed industry to imitate the acorn profile by adding unsaturated fats in the pig’s diet [14,17]. This caution when publishing information about the Iberian pig feeding regime has been pointed out in order to prevent fraud [15]. In this sense, the use of a non-targeted approach such as fingerprinting could offer an important advantage in comparison to a targeted one since no information about specific compounds is published. Furthermore, the use of different expensive equipment with higher resolution would be recommendable in order to clearly identify specific compounds.

### 4.3. General Discussion

Although many studies discriminating or authenticating the Iberian pig feeding regime can be found in the literature, these are usually focused on the evaluation of the current commercial categories which are defined by the regulation [5]. In this sense, research is frequently published without considering specific practices from the complex sector of the Iberian pig, such as *postre* or feed supplementation during the finishing phase of acorn-fed animals, which is clearly presented in the current study for the first time.

The high classification rates obtained in this work after GC–IMS analysis and data treatment were in accordance with previous studies using this type of methodology to evaluate volatilome analysis as a feasible tool to authenticate Iberian pig feeding regime. However, results about the Iberian pig production and feeding regime are usually obtained after the analysis of samples such as raw subcutaneous fat [23,24,25,26], raw meat [27,28,29,30] or even hepatic tissue [31], all usually collected at the slaughterhouse. On the other hand, dry cured ham and fat obtained after the curing and ripening process have been widely employed too [32,33,34,35]. Therefore, the results here presented are interesting not only due to the consideration for the first time of the fraudulent practice of *postre* (feed supplementation) during the final fattening phase of the acorn-fed Iberian pig in the *dehesa*, but also to the use of faeces as an innovative biological sample. Faeces constitute a complex biological sample, which provides useful information about the individual allowing an easy collection [12]. This alternative sample offers an important advantage for the evaluation of the Iberian pig feeding regime in comparison to the latter, as it allows in vivo analysis and facilitates field inspections. In other words, while the usual samples used in this field are collected at the slaughterhouse (*postmortem*), the use of faeces does not require the slaughter of pigs. Therefore, the sample collection and the subsequent analysis could be *antemortem* procedures, and this would allow the control of the acorn-fed Iberian pig diet directly at farms.

These results highlight the capacity of GC–IMS to detect the fraudulent use of supplementary feed or *postre* during the fattening of the acorn-fed Iberian in the *dehesa* and its discrimination from the grazing and extensive diet that these animals are supposed to have. Therefore, this finding could trigger a discussion about the possible categorization of this feeding strategy, in order to distinguish it from the exclusive feeding of natural resources and to clarify the market. In this sense, there are some authors who have suggested a modification of Iberian product categories in order to simplify the market [36]. This is consequence of the consumers’ greater demand of complete information from the products, whose selection is nowadays frequently difficulted due to an ambiguous labelling or origin information. Despite the good results achieved in this preliminary discrimination of the use of feed supplementation in the acorn-fed Iberian pig diet, further research is required on this issue. The continuous collaboration with farmers who give reliable information about the animals’ feed, which may be hard due to the controversy of this practice, is considered essential.

## 5. Conclusions

The proposed methodology based on GC–IMS for volatilome analysis has shown excellent results for the detection of feed supplementation during the last stage of acorn-fed Iberian pig fattening. Considering the classification success achieved, volatilome information can show differences between animals fed exclusively with natural resources in the *dehesa* and those also supplemented with feed. The two chemometric approaches evaluated (fingerprinting and individual features) have been shown to be useful for GC–IMS data treatment, obtaining classification rates above 85%. This study constitutes the first approximation of the use of an analytical methodology to evaluate the fraudulent feed supplementation during the final fattening of the acorn-fed Iberian pig in the *dehesa*. The results here presented highlight the potential of faeces volatilome analysis through GC–IMS to authenticate the extensive diet that the acorn-fed Iberian pig is supposed to have along the *montanera* period.

This methodology to authenticate pasture-raised pigs could be useful and adapted for other upstanding livestock production systems such as organic farming, or to distinguish natural diets in field; all this interest is in a social context where there is market competition between farmed and wild animal products, and a grazing diet is an important demand and a consumer preference.

## Figures and Tables

**Figure 1 animals-13-00226-f001:**
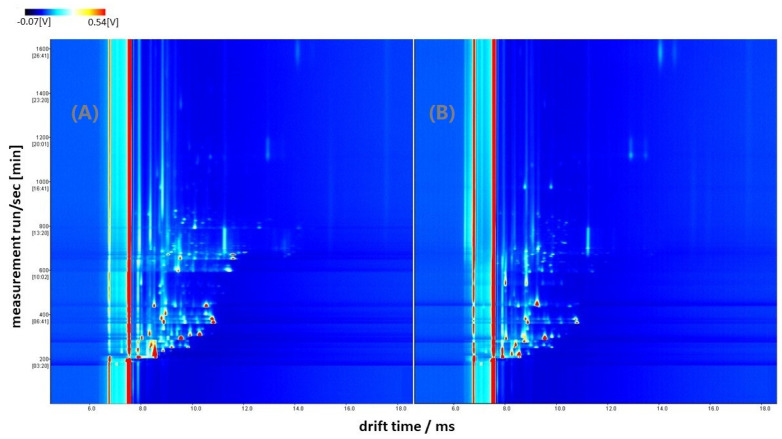
Topographic plots of GC–IMS spectra obtained for acorn-fed (**A**) and supplementation (**B**) samples.

**Figure 2 animals-13-00226-f002:**
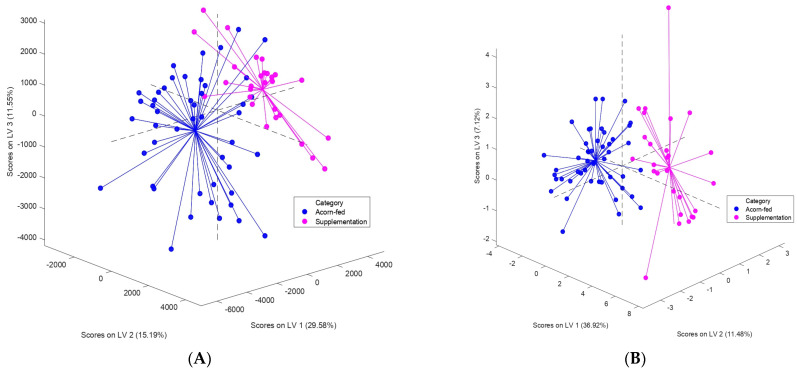
Score plots of PLS-DA models performed: (**A**) Model 1: non-targeted (spectral fingerprint) approach; (**B**) Model 2: targeted (individual features) approach (21 features).

**Figure 3 animals-13-00226-f003:**
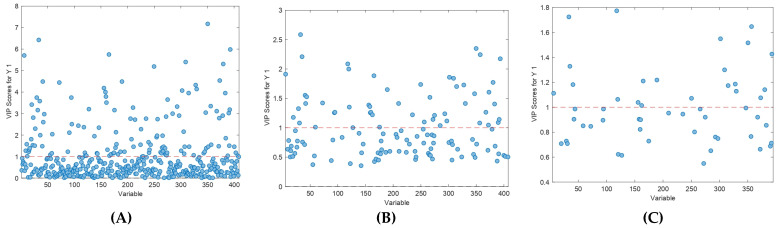
Consecutive VIP score evaluation performed for feature reduction in Model 2 (targeted approach), repeated three times (**A**–**C**) and eliminating features with a VIP value < 1.

**Figure 4 animals-13-00226-f004:**
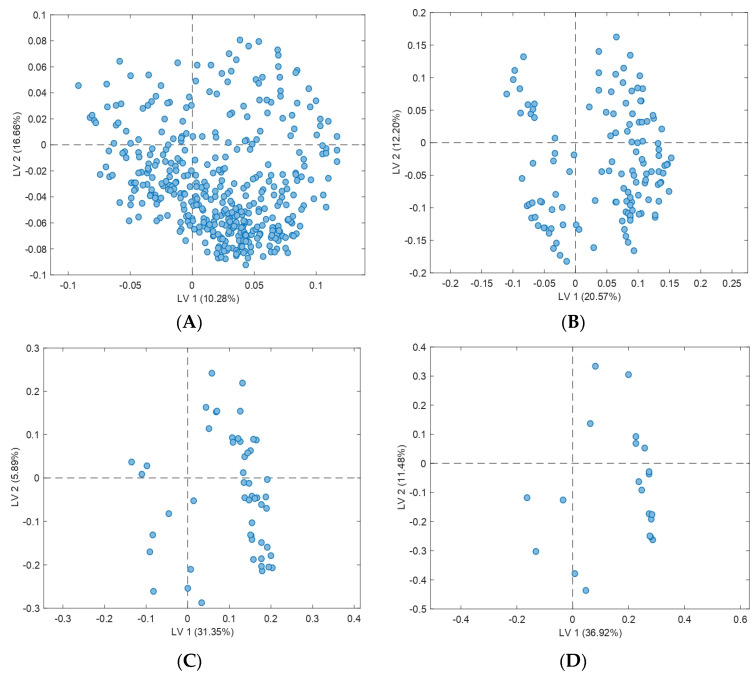
Loading plots obtained for consecutive PLS-DA models performed through the reduction of the initial 409 features in the targeted approach (Model 2) ((**A**–**D**): performed with all the 409 initial features–performed only with the 21 final features).

**Table 1 animals-13-00226-t001:** External validation results (confusion matrix) obtained by the PLS-DA model (Model 1) using blind samples for the non-targeted approach (spectral fingerprint).

		**Real Classification**	
		Acorn-Fed	Supplementation
**Model prediction**	Acorn-fed	11	0
	Supplementation	3	8
	Total	14	8
	% category classification	78.6	100.0
	% total classification	86.4	

**Table 2 animals-13-00226-t002:** External validation results (confusion matrix) obtained by the PLS-DA model (Model 2) using blind samples for the targeted approach (individual features).

		**Real Classification**	
		Acorn-Fed	Supplementation
**Model prediction**	Acorn-fed	14	0
	Supplementation	0	8
	Total	14	8
	% category classification	100.0	100.0
	% total classification	100.0	

**Table 3 animals-13-00226-t003:** Classification error, sensitivity and specificity percentages obtained for classification models performed.

Model	Category	Calibration			Cross-Validation			Prediction		
		Class. Error	SE	SP	Class. Error	SE	SP	Class. Error	SE	SP
1—non-targeted (Spectral fingerprint)	Acorn-fed	15.2	84.8	95.0	18.2	81.8	75.0	21.4	78.6	100.0
	Supplementation	5.0	95.0	84.8	25.0	75.0	81.8	0.0	100.0	78.6
2—targeted (Individual features)	Acorn-fed	3.0	97.0	95.0	3.0	97.0	95.0	0.0	100.0	100.0
	Supplementation	5.0	95.0	97.0	5.0	95.0	97.0	0.0	100.0	100.0

Class: classification. SE: sensitivity. SP: specificity.

**Table 4 animals-13-00226-t004:** Results obtained for the Student t test performed with the 21 final features of PLS-DA model (targeted approach).

Feature Number	Acorn-Fed		Supplementation		*p*	Feature Number	Acorn-Fed		Supplementation		*p*
	Intensity *	Standard Deviation *	Intensity	Standard Deviation *			Intensity *	Standard Deviation *	Intensity *	Standard Deviation *	
6 ‡	104.08	31.54	165.05	56.89	<0.001	302 †	2.59	1.41	1.53	0.51	<0.001
33 ‡	68.83	30.44	118.99	36.15	<0.001	309 ‡	9.85	2.59	13.46	4.73	0.001
35 ‡	100.49	46.13	138.39	49.21	0.001	315 †	4.82	3.12	2.58	1.52	<0.001
41 ‡	37.26	19.35	66.74	35.84	<0.001	328 ‡	75.58	36.92	164.57	117.81	0.001
118 ‡	8.89	4.73	12.67	8.67	0.017	330 ‡	11.81	5.90	22.30	13.82	0.001
120 †	4.28	2.19	3.67	2.00	0.218	350 ‡	4.70	1.91	9.24	4.35	<0.001
156 ‡	40.61	18.62	119.72	106.06	0.001	357 †	88.41	22.01	80.88	21.00	0.149
162 †	12.11	9.10	8.41	5.59	0.056	373 ‡	18.44	10.10	52.08	38.69	<0.001
165 ‡	26.67	11.98	60.09	40.10	<0.001	380 ‡	21.66	9.63	38.48	20.67	<0.001
189 ‡	22.59	13.64	52.68	41.50	0.001	393 ‡	23.51	8.07	53.26	29.99	<0.001
250 ‡	3.29	1.35	5.33	2.11	<0.001						

* Data presented correspond to the mean results of each group. † Features with a higher intensity detected in the acorn-fed group. ‡ Features with a higher intensity detected in the supplementation group.

## Data Availability

The data supporting the findings of this study are available upon request from the author Pablo Rodríguez-Hernández (v22rohep@uco.es) and with permission from farmers included in the study.

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
