# Peer review of "Feed Supplementation Detection during the Last Productive Stage of the Acorn-Fed Iberian Pig through a Faecal Volatilome Analysis"

_animals, 2023, doi:10.3390/ani13020226_

Round 1

Reviewer 1 Report

A methodology to detect the fraudulent use of feed for acornfed Iberian pig’s fattening is necessary. GC-IMS is useful. The results provide new insight into feed supplementation detection in acorn-fed Iberian pig, however, there are still several comments should be taken into consideration:

1.  Was all feed supplementation same in different farm? Further explanations about the supplementation diet should be explained. The diet received plays a critical role in the faeces. Therefore showing the characteristics of the diet is critical.

2.Also it is necessary to report the initial body weight and the final body weight of the animals.

3. Were age and body weight different between two groups? This is very important informatin in this study.

Author Response

We would like to thank reviewer 1 for the changes suggested.

According to the reviewers’ suggestions, several changes have been included in the manuscript. These are marked with red (and strike out) and green colors.

Reviewer 2 Report

The manuscript deals with Feed supplementation detection during the last productive stage of the acorn-fed Iberian pig through a faecal volatilome analysis. The manuscript provides an interesting approach for a field quality control of pigs fed different feed matrices.

Hence the manuscript fits in the scope of the journal. However, there are some remarks which have to be considered before the paper can be accepted for publication.

A major weakness is the missing information in the material and methods part:

How long was the adaption time of the feed until sampling of the faeces started?

The authors should include the sex of the animals in the statistical model (two factorial design as well as their interaction) or at least discuss a potential effect of the sex in the discussion.

More detailed information regarding the applied feed supplementation would be helpful to improve the discussion. Perhaps the different supplementation feeds can be included as co-variable in the statistical model to improve the statistical power of this study.

A table (including number of male and female pigs per treatment) explaining point 2.1 would be very helpful for an easier understanding of the study design.

The heading of the figures should be placed below.

Figure 1: A more precise labeling of this figure would be appropriate

Author Response

We would like to thank reviewer 2 for the changes suggested.

According to the reviewers’ suggestions, several changes have been included in the manuscript. These are marked with red (and strike out) and green colors.

Round 2

Reviewer 1 Report

 This manuscript could be accpted.

Reviewer 2 Report

The manuscript is now moderatly improved. Hence I have no more comments.